# Genetic Diversity and Population Structure of Tufted Deer (*Elaphodus cephalophus*) in Chongqing, China

**DOI:** 10.3390/ani15152254

**Published:** 2025-07-31

**Authors:** Fuli Wang, Chengzhong Yang, Yalin Xiong, Qian Xiang, Xiaojuan Cui, Jianjun Peng

**Affiliations:** 1School of Life and Health Sciences, Hunan University of Science and Technology, Xiangtan 411201, China; 23010901009@mail.hnust.edu.cn; 2Animal Biology Key Laboratory of Chongqing Education Commission of China, College of Life Sciences, Chongqing Normal University, Chongqing 401331, China; drczyang@126.com (C.Y.); ylzbl2333@163.com (Y.X.); xq17815366557@163.com (Q.X.)

**Keywords:** tufted deer, mitochondrial DNA, genetic diversity, Chongqing

## Abstract

The tufted deer (*Elaphodus cephalophus*) is a rare and endangered animal that lives only in China and Myanmar, and protecting it requires understanding how healthy and connected its populations are. This study looked at three groups of tufted deer living in different mountains in China to see how genetically diverse they are and the extent of gene flow among them. By analyzing specific parts of their DNA, researchers found that one area, Simian Mountain, has the most genetic diversity, making it very important for the species’ survival and as a source of genetic exchange. The study also revealed that one group, Jinfo Mountain, is more isolated, likely due to its unique terrain, which could hinder its ability to mingle with other groups. Evidence suggests that the overall population underwent a significant expansion during the Pleistocene epoch, but some groups remain separated. These findings highlight the importance of conserving and managing these deer populations carefully, especially by protecting key areas like Simian Mountain and establishing ecological corridors, particularly to enhance connectivity between the Jinfo and Simian Mountain populations. The results will help scientists and conservationists develop effective strategies to safeguard the tufted deer and ensure the future of this unique species.

## 1. Introduction

The tufted deer (*Elaphodus cephalophus*), the sole species within the genus *Elaphodus* of the Cervidae family, is classified as Near-Threatened (NT) by the International Union for Conservation of Nature (IUCN) Red List of Threatened Species [1]. Characterized by its yellowish-brown coat, prominent brown frontal tuft, and short, unbranched antlers [2], the tufted deer possesses unique morphological features that underscore its phylogenetic significance. The endemic species is primarily distributed in China, with its southern edge extending into Myanmar [1,2]. Chongqing, situated on the southeastern rim of the Sichuan Basin, features a landscape dominated by mountains and hills [3]. This distinctive environment provides an ideal habitat for the tufted deer, facilitating the establishment of stable populations within the region.

The unique genetic background of tufted deer makes the study of its genetic diversity highly valuable for developing conservation strategies, understanding population differentiation, and exploring adaptive evolution. Analyzing microsatellite markers, mitochondrial DNA, or whole-genome data allows for the assessment of population genetic structure, demographic history, and environmental adaptability [4,5,6,7]. This information is crucial for comprehending current population status and formulating precise conservation strategies aimed at promoting gene flow and preserving genetic resources. From an ecological and evolutionary perspective, genetic diversity reflects a species’ health, adaptation mechanisms, and evolutionary history, helping to decipher the resilience of tufted deer to environmental changes [8,9,10]. Furthermore, high levels of genetic diversity enhance a species’ environmental adaptability and disease resistance, providing vital indicators for species classification and ecosystem health monitoring [11,12,13].

Mitochondrial DNA (mtDNA) exists in multiple copies per cell, enabling its efficient amplification via Polymerase Chain Reaction (PCR) even from samples with low DNA concentration or significant degradation [14]. Furthermore, mtDNA is predominantly maternally inherited and lacks sexual recombination, conferring unique advantages for investigating population genetic structure, phylogeny, and demographic history [9,15,16]. Specifically, the cytochrome b (Cyt b) gene, a protein-coding region, evolves at a moderate rate [9]. This allows it to retain sufficient variability for detecting genetic differentiation and long-term divergence among populations while remaining relatively conserved [17]. Conversely, the D-loop region, a highly variable non-coding control region, is instrumental in revealing fine-scale genetic structure, haplotype distribution, and recent population dynamics within populations [18,19,20]. The combined analysis of these two markers leverages their complementary strengths, enhancing the reliability of genetic diversity assessments and yielding a more comprehensive and accurate understanding of population history, structure, and dynamic changes [21,22,23].

Despite the tufted deer being found only in parts of China and Myanmar and possessing a unique genetic background, systematic research into its genetic diversity and population structure remains limited. A 2016 study utilized mtDNA control region (CR) and nuclear microsatellite markers to analyze the genetic structure and gene flow between two tufted deer populations in Bashan and Wuling Mountains, separated by the natural barrier of the Yangtze River [24]. Therefore, in this study, we collected fecal samples from three representative wild populations of tufted deer in Chongqing—Jinfo Mountain, Simian Mountain, and the Northeastern Mountainous region. By amplifying and analyzing mitochondrial Cyt b and D-loop gene sequences, we systematically evaluated the genetic diversity, population structure, gene flow, and demographic history of these populations. The specific objectives are (1) to assess the genetic diversity levels of tufted deer populations in different regions of Chongqing; (2) to reveal the genetic structure among these populations; (3) to analyze the extent of gene flow between populations; and (4) to reconstruct the demographic history of the main populations. The results of this study will provide a theoretical basis and data support for the conservation and scientific management of tufted deer genetic resources.

## 2. Materials and Methods

### 2.1. Sample Collection and DNA Extraction

To minimize disturbance to the target species, this study employed a non-invasive fecal sampling approach to obtain genetic material from tufted deer. Host DNA was extracted from collected tufted deer fecal samples [25,26,27,28]. Between 2022 and 2023, a total of 46 tufted deer fecal samples were collected from distinct forest regions within Chongqing, China, using a systematic opportunistic sampling protocol with strict spatial separation criteria: Jinfo Mountain National Nature Reserve (JF, *n* = 13); Simian Mountain National Nature Reserve (SM, *n* = 21); and the Northeastern Mountainous areas of Chongqing (NEM, *n* = 12) (Figure 1). In this study, all samples were assigned to three geographical populations: Simianshan (SM), Jinfoshan (JF), and Northeastern Chongqing (NEM). This delimitation was based on a priori criteria including geographical location, topographical barriers, and ecological connectivity. Although the SM and JF populations are geographically proximate, the Jinfoshan area was treated as a highly isolated ecological unit due to its unique high altitude, steep terrain, and significant habitat fragmentation resulting from tourism development [29]. In contrast, despite spanning a wider geographical area, all sampling sites for the NEM population are located within the topographically continuous Daba Mountain Range, which lacks major internal geographical barriers, and were thus considered a single metapopulation with potential internal gene flow. The wild sampling protocol required (1) collection of fresh samples to represent recent defecation events and (2) a minimum straight-line distance of 500 m between any two sampling sites [30]. During sampling, personnel wore sterile disposable gloves, and collected samples were immediately stored at −80 °C.

Host DNA was extracted from fecal samples using the QIAamp DNA Stool Mini Kit (QIAGEN, Hilden, Germany). Subsequently, DNA concentration and purity were measured on a Nano-400A Micro-Spectrophotometer (Aosheng Instrument Co., Ltd., Hangzhou, China) before storage at −20 °C for later use.

### 2.2. PCR Amplification, Gene Cloning, and Sequencing

For the amplification of mtDNA, Polymerase Chain Reaction (PCR) was employed with specific primer pairs (Table 1). PCRs were performed using TIANGEN’s 2× Taq PCR MasterMix II (TIANGEN BIOTECH, Beijing, China). The thermocycling program consisted of an initial denaturation at 94 °C for 3 min, followed by 35 cycles of 94 °C for 30 s (denaturation), 57 °C for 30 s (annealing), and 72 °C for 1 min (extension). A final extension step was carried out at 72 °C for 5 min.

PCR products were purified using the Biospin Gel Extraction Kit and subsequently cloned into the pTOPO-TA/Blunt Cloning Kit (Aidlab Biotechnologies Co., Ltd., Beijing, China). Individual colonies were inoculated into 400 μL of Luria–Bertani (LB) medium containing 100 μg/mL ampicillin (Amp) and incubated at 37 °C with shaking for 2 h. Colony PCR was then performed to identify positive clones, and samples were sent to Bioscience Company for sequencing (Changsha, China).

### 2.3. Data Analysis

After obtaining sequencing results, bases were manually verified against electropherograms to ensure accuracy. Sequences were then compared with entries in NCBI BLAST (https://blast.ncbi.nlm.nih.gov/Blast.cgi, accessed on 25 February 2025) [31] to confirm their derivation from tufted deer. MEGA 11.0.13 software was used for multiple sequence alignment and trimming, as well as for calculating haplotype genetic distances and nucleotide composition [32]. To analyze population genetic structure, DnaSP 6.12.03 was employed to compute nucleotide diversity (Pi), number of haplotypes (H), haplotype diversity (Hd), and average nucleotide differences (K), along with mismatch distribution analysis [33]. Analysis of molecular variance (AMOVA) and genetic differentiation indices (F*_st_*) were obtained using Arlequin 3.5 [34]. Neutrality tests, including Tajima’s D and Fu’s FS, were also conducted. Gene flow (*N*_m_) was calculated using the formula *N*_m_ = 0.25 (1 − F*_st_*)/F*_st_* [35]. A median-joining haplotype network was constructed with PopART 1.7 [36,37]. Mitochondrial genome data, obtained from NCBI (Appendix A), were analyzed using Bayesian inference as implemented in BEAST v2.6.7 to generate a time-calibrated phylogenetic tree and a Bayesian Skyline Plot detailing the species’ demographic history [38]. As no specific mutation rate has been published for the tufted deer, we used the rate from a closely related species, the Siberian roe deer (*Capreolus pygargus*), setting the mitochondrial mutation rate to 8.91 × 10^−9^ substitutions/site/year for time calibration [39].

## 3. Results

### 3.1. Sequence Characteristics and Genetic Diversity

Mitochondrial DNA analysis of three muntjac populations (JF/SM/NEM) from Chongqing, China, revealed distinct sequence characteristics. The Cyt b gene exhibited a notable AT bias (AT = 58.4%, GC = 41.6%) (Table 2). In contrast, the D-loop region displayed a more balanced nucleotide composition (AT = 51.7%, GC = 48.3%) (Table 2). The AT-rich patterns observed across both sequences align with the typical evolutionary trajectory of mammalian mitochondrial DNA, consistent with established mitochondrial gene characteristics.

Genetic variation in the D-loop region was markedly higher than in the Cyt b gene, as evidenced by three key indices: (1) a greater number of polymorphic sites (D-loop S = 43 vs. Cyt b S = 17); (2) higher nucleotide diversity (π = 0.02127 vs. 0.01424); and (3) richer haplotype diversity (Hd = 0.97295 vs. 0.96522) (Table 3).

Population-specific analyses revealed the following patterns: (1) For the Cyt b gene, the NEM population exhibited the highest genetic diversity (Hd = 0.96970; π = 0.01602), suggesting a historically large effective population size or greater demographic stability. (2) For the D-loop region, the SM population displayed the highest diversity (Hd = 0.97143; π = 0.01982), implying either more extensive gene flow or enhanced habitat connectivity. (3) The JF population showed the lowest diversity at both markers (Cyt b Hd = 0.87179; D-loop Hd = 0.91026), which may indicate prolonged genetic isolation or historical bottleneck events.

### 3.2. Phylogenetic Relationships and Genetic Structure

Analysis of the Cyt b gene region identified a total of 25 unique haplotypes across all samples. Within the populations, there were 6 haplotypes in JF, 12 in SM, and 10 in NEM. Notably, Hap_11 was the most common haplotype, appearing frequently across samples. The D-loop region revealed a total of 30 haplotypes, with 9 in JF, 16 in SM, and 10 in NEM. Hap_4 was a predominant haplotype shared among multiple populations (Appendix A).

The time-calibrated Bayesian phylogenetic trees revealed deep evolutionary divergences dating back millions of years. The deepest split among the haplotypes occurred approximately 4.9 million years ago (Mya) during the Pliocene epoch (Figure 2). Many subsequent diversification events took place throughout the Pleistocene, indicating a long and complex evolutionary history rather than a single recent expansion. While no strictly geographically exclusive clades were formed among the three sampled populations, suggesting some historical gene flow, the JF population’s haplotypes tend to cluster in more recent, shallower parts of the tree.

This is further supported by the haplotype networks, where the high-diversity SM population occupies a central, hub-like position, connecting numerous low-frequency haplotypes, characteristic of a source or ancestral population (Figure 3).

### 3.3. Genetic Differentiation and Gene Flow

The mitochondrial Cyt b gene and D-loop region consistently indicated the highest genetic differentiation for the JF population relative to the SM and NEM populations (Cyt b F*_st_*: JF-SM = 0.27405, JF-NEM = 0.25385; D-loop F*_st_*: JF-SM = 0.23394, JF-NEM = 0.26201). This pronounced differentiation suggests significant geographical isolation affecting the JF population. In contrast, genetic differentiation between the SM and NEM populations was notably lower (Cyt b F*_st_* = 0.09604; D-loop F*_st_* = 0.06227), implying higher genetic connectivity between these two groups (Table 4 and Table 5).

Gene flow (*N*_m_) analyses showed notable disparities between the two markers. The D-loop region consistently detected higher gene flow intensities compared to the Cyt b region, particularly between the SM and NEM populations (D-loop *N*_m_ = 3.76494 vs. Cyt b *N*_m_ = 2.35307). This discrepancy reflects the D-loop’s greater sensitivity to more recent gene exchange events. Intriguingly, the SM population appears to act as a gene flow “hub,” exhibiting a noticeably higher unidirectional gene flow towards the JF population (Cyt b *N*_m_ = 0.66218; D-loop *N*_m_ = 0.81883) than in the reverse direction.

### 3.4. Demographic History

The demographic history of the tufted deer populations was investigated using a combination of Bayesian Skyline Plots (BSPs), neutrality tests, and mismatch distribution analysis, revealing a complex demographic history characterized by long-term stability overlaid by a past expansion event.

The BSP analyses for both Cyt b and D-loop datasets provided a consistent picture of the species’ long-term demographic history, indicating a long period of stability followed by a gradual but significant increase in effective population size that began during the Middle to Late Pleistocene (approximately 1.0 Mya to 0.5 Mya) (Figure 4). This signal of historical expansion is consistent with the diversification patterns observed in the phylogenetic trees.

Neutrality tests and mismatch distribution analyses, which reflect broader historical timescales, provided additional context. For the faster-evolving D-loop region, the SM population showed a highly significant negative Fu’s Fs value (−5.73756, *p* = 0.004), a strong indicator of population expansion due to an excess of new mutations. The mismatch distribution for the D-loop was a classic unimodal curve (Figure 5b), which is also consistent with a past expansion event. In contrast, the more conserved Cyt b marker showed non-significant Tajima’s D values and a ragged, multimodal mismatch distribution (Figure 5a), a pattern more typical of a demographically stable population, rather than simple expansion. The discrepancy between Tajima’s D (often positive) and Fu’s Fs (often negative) across populations (Table 6) further suggests a complex history where signals of older stability are overlaid with signals of more recent Pleistocene expansion.

## 4. Discussion

Our study reveals a demographic history shaped by deep time and major climatic events. The deep divergence (~4.9 Mya) and subsequent diversification throughout the Pleistocene suggest that the genetic structure of tufted deer was primarily forged by long-term evolutionary processes, not recent events. The high genetic differentiation of the Jinfo (JF) population is better explained by long-term isolation, potentially in a glacial refugium during Pleistocene climate oscillations [40], rather than a recent founder event. This is supported by highly asymmetrical gene flow (*N*_m_, Table 4 and Table 5) and the population’s distinct position in the haplotype network.

The dominant demographic signal in our data is a significant population expansion that began in the Middle to Late Pleistocene, as detected by Bayesian Skyline Plot (BSP), Fu’s Fs tests, and mismatch distribution analysis. This historical expansion is strongly coupled with known climatic drivers, such as the glacial–interglacial cycles of the Pleistocene, which are known to have profoundly impacted the distribution and demography of many East Asian fauna.

Mt DNA analysis confirms high gene flow (*N*_m_ > 2.35) between SM and NEM populations, indicating the Yangtze River is not an absolute barrier, consistent with Sun et al. (2016) [24]. In contrast, the JF population is highly isolated (F*_st_* > 0.23), likely due to its unique topography [29], which creates significant physical barriers to dispersal, highlighting its conservation value but also its vulnerability to further fragmentation. However, this matrilineal pattern conflicts with the expected male-biased dispersal in cervids [41,42,43]. We hypothesize that assortative mating—where local females prefer local males—maintains maternal lineage integrity despite potential male influx. This preference may not be for inbreeding avoidance [43]. Therefore, the observed genetic structure likely results from an interplay of female philopatry, hypothesized assortative mating preferences, and geographic barriers, which warrants further verification with nuclear markers.

This study has several limitations that should be acknowledged. Methodologically, our non-invasive sampling with 500 m spatial separation, while a standard approach, cannot entirely exclude resampling the same individual, and it resulted in unbalanced sample sizes (SM = 21, NEM = 12, JF = 13) that require cautious interpretation of genetic diversity estimates, which could be skewed by sampling effort. Although the incongruence between markers (D-loop vs. Cyt b) suggests our findings are not mere sampling artifacts, this issue highlights the need for larger, more balanced samples for robust assessment. More fundamentally, our reliance on mtDNA provides only a maternal perspective, omitting paternal gene flow and nuclear genomic effects. Therefore, future research incorporating nuclear markers (e.g., microsatellites, SNPs) is essential for a comprehensive, biparental understanding of the population’s genetic structure and to definitively identify individuals, thereby strengthening conservation strategies.

Based on these findings, future work will combine habitat modeling and landscape analysis to identify and assess potential ecological corridors for tufted deer in Chongqing, providing a scientific basis for improving connectivity, particularly for the isolated Jinfo Mountain population. Our findings provide key conservation insights based on evolutionary history. First, the Jinfo (JF) population, with its unique genetic signature from long-term isolation, warrants designation as a distinct Evolutionarily Significant Unit (ESU) or Management Unit (MU) requiring targeted local conservation. Second, the SM/NEM metapopulation represents the historical genetic ‘core’ for the species in this region. Protecting its habitat integrity and connectivity is paramount for regional persistence. Therefore, we recommend a dual strategy: managing the isolated JF population separately while safeguarding the central core metapopulation.

## 5. Conclusions

This research comprehensively evaluated the genetic diversity and population structure of tufted deer in the Chongqing area by integrating analyses of the mitochondrial Cyt b gene and D-loop regions. Our findings reveal not only rich genetic variation and a distinct population structure but also a deep evolutionary history shaped by Pleistocene climate change. High genetic diversity is predominantly concentrated in the Simian Mountain area, establishing it as the historical genetic ‘core’ for the species in this region. In contrast, the Jinfo Mountain population exhibits profound genetic isolation, which we interpret as the result of long-term separation, likely within a glacial refugium, warranting its designation as a distinct Management Unit (MU) or Evolutionarily Significant Unit (ESU). The primary demographic signal is a significant population expansion during the Pleistocene, highlighting the influence of past climatic events on the species’ trajectory. Therefore, we recommend a dual conservation strategy: implementing targeted management for the evolutionarily unique Jinfo population and constructing ecological corridors to enhance connectivity with the core metapopulation, ensuring the long-term persistence and evolutionary potential of tufted deer.

## Figures and Tables

**Figure 1 animals-15-02254-f001:**
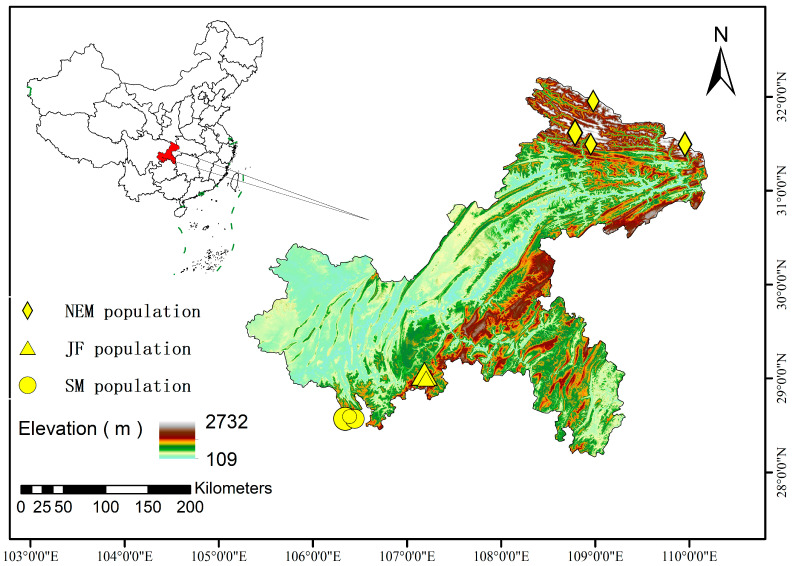
Collection sites of tufted deer fecal samples.

**Figure 2 animals-15-02254-f002:**
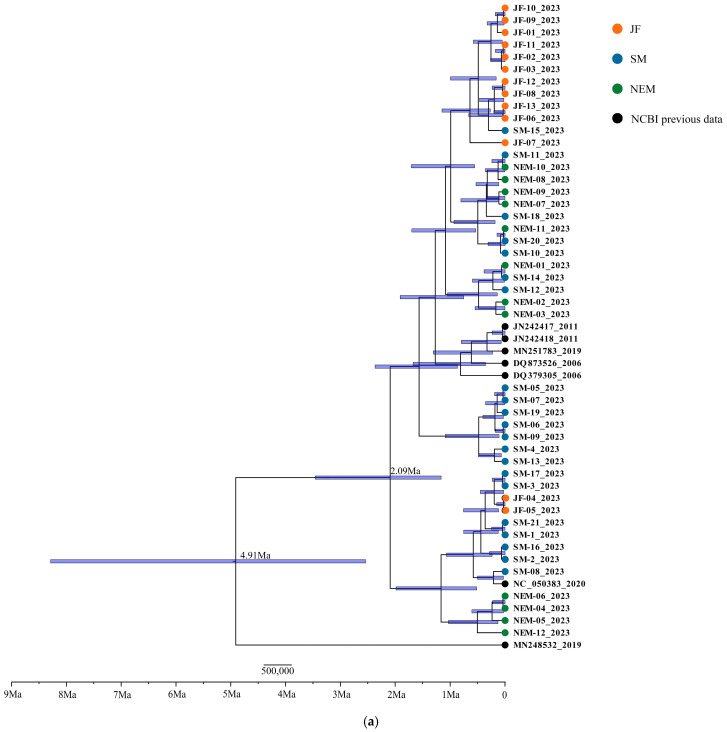
Time-calibrated Bayesian trees of haplotypes derived from Cyt b gene (**a**) and D-loop region (**b**).

**Figure 3 animals-15-02254-f003:**
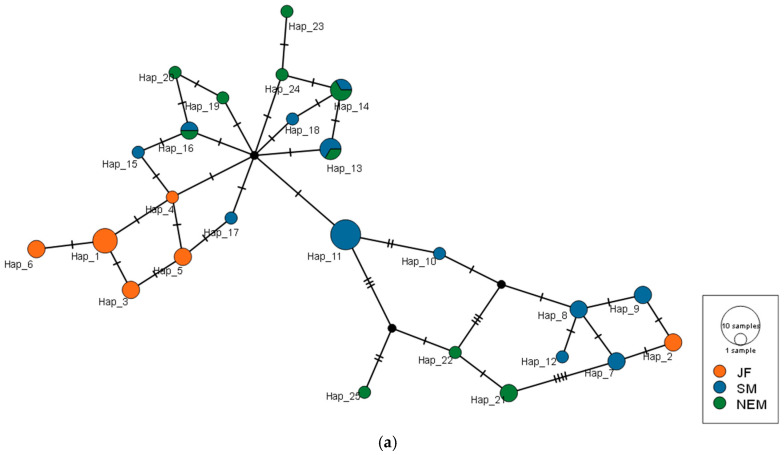
TCS haplotype networks for the tufted deer based on the mitochondrial Cyt b gene (**a**) and D-loop region (**b**). The size of each circle is proportional to its frequency. Each hash mark on the lines represents a single mutational step, and small black dots indicate inferred median vectors.

**Figure 4 animals-15-02254-f004:**
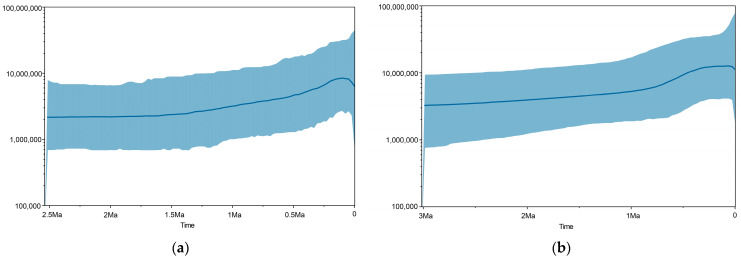
Bayesian Skyline Plot (BSP) analysis of tufted deer populations: evidence for recent demographic expansion from (**a**) Cyt b and (**b**) D-loop sequence data (The solid line represents the median estimate of the effective population size (Ne), while the shaded area corresponds to the 95% Highest Posterior Density (HPD) interva).

**Figure 5 animals-15-02254-f005:**
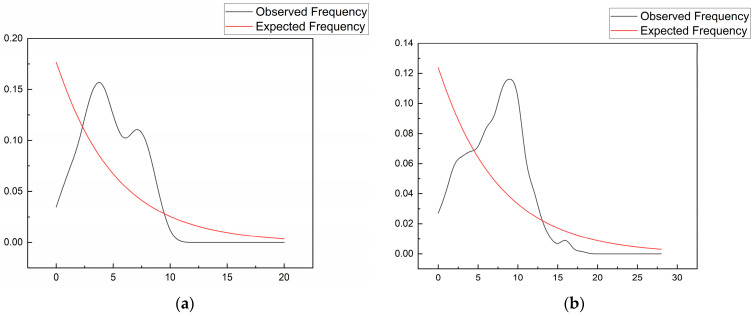
Observed versus expected mismatch distribution curves for tufted deer populations based on (**a**) Cyt b and (**b**) D-loop sequences.

**Table 1 animals-15-02254-t001:** Primer sequences used for PCR amplification.

Primer Name	Primer Sequence (5′-3′)	Product Length
Cyt b	F: 5′-CAAACGGAGCATCAATGTT-3′	359 bp
	R: 5′-TGTCTCGTGGAGAAAGAGT-3′	
D-loop	F: 5′-TAAGTCAAATCAGTCCTCGTCAA-3′	369 bp
	R: 5′-GTTAAGTCCAGCTACAATTCATG-3′	

**Table 2 animals-15-02254-t002:** Nucleotide composition of Cyt b gene and D-loop region in muntjac populations.

Items	Base Composition (%)						
	T(U)	C	A	G	AT(U)	GC	Total
Cyt b	30.7	24.5	27.7	17.1	58.4	41.6	327
D-loop	28.1	27.5	23.6	20.8	51.7	48.3	334

**Table 3 animals-15-02254-t003:** Genetic diversity indices of the Cyt b gene and D-loop region in tufted deer.

Genetic Marker	Group	Polymorphism Site Analysis (S)	Number of Haplotypes (H)	Haplotype Diversity (Hd)	Average Number of Nucleotide Differences (K)	Nucleotide Diversity (π)
Cyt b	JF	8	6	0.87179	2.74359	0.00839
	SM	12	12	0.90952	4.04762	0.01238
	NEM	13	10	0.96970	5.16667	0.01602
	Total Data Estimates	17	25	0.96522	4.65700	0.01424
D-loop	JF	18	9	0.91026	3.92308	0.01178
	SM	25	16	0.97143	6.60000	0.01982
	NEM	20	10	0.95455	7.96970	0.02393
	Total Data Estimates	43	30	0.97295	7.08213	0.02127

**Table 4 animals-15-02254-t004:** Genetic differentiation (F*_st_*, upper right) and gene flow (*N*_m_, lower left) among tufted deer populations based on Cyt b sequences.

	F*_st_*	JF	SM	NEM
*N* _m_				
JF			0.27405	0.25385
SM		0.66218		0.09604
NEM		0.73482	2.35307	

**Table 5 animals-15-02254-t005:** Genetic differentiation (F*_st_*, upper right) and gene flow (*N*_m_, lower left) among tufted deer populations based on D-loop sequences.

	F*_st_*	JF	SM	NEM
*N* _m_				
JF			0.23394	0.26201
SM		0.81883		0.06227
NEM		0.70417	3.76494	

**Table 6 animals-15-02254-t006:** Tajima’s D and Fu’s FS values for tufted deer populations based on Cyt b and D-loop sequences.

Genetic Marker	Group	Tajima’s D	Tajima’s D *p*-Value	FS	FS *p*-Value
Cyt b	JF	0.24882	0.63800	−0.31705	0.44100
	SM	0.75692	0.78000	−2.89506	0.08800
	NEM	0.84804	0.84100	−3.40205	0.03600
	Mean	0.61793	0.75300	−2.20472	0.18833
	Standard Deviation	0.32289	0.10416	1.65431	0.22036
D-loop	JF	−1.37337	0.08100	−2.55315	0.07100
	SM	−0.19214	0.48100	−5.73756	0.00400
	NEM	0.89333	0.86300	−2.08194	0.13900
	Mean	−0.22406	0.47500	−3.45755	0.07133
	Standard Deviation	1.13369	0.39103	1.98855	0.06750

## Data Availability

The original contributions presented in this study are included in this article. Further inquiries can be directed to the corresponding author(s).

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
