# Peer review of "Genetic Diversity and Population Structure of Tufted Deer (Elaphodus cephalophus) in Chongqing, China"

_animals, 2025, doi:10.3390/ani15152254_

Round 1
Reviewer 1 Report
Comments and Suggestions for Authors
This article is important as it examines the genetic diversity and connectivity of tufted deer populations in China, which is useful for conservation efforts. However, several corrections and improvements are needed, such as adding more background information, improving the explanation of the methods and results, and including additional important details to strengthen the study.

Reviewer 2 Report
Comments and Suggestions for Authors
The article by Fuli Wang, Chengzhong Yang, Yalin Xiong, Qian Xiang, Xiaojuan Cui and Jianjun Peng "Genetic Diversity and Population Structure of tufted deer (Elaphodus cephalophus) in Chongqing, China" is devoted to an interesting issue of population genetics related to the study of the population structure of an endemic ungulate species. The tufted deer (Elaphodus cephalophus) is a rare and endangered animal that lives only in Southeast Asia. This fact gives the article high relevance and importance from an environmental point of view. The article describes the genetic structure of three subdivided populations of E. cephalophus using two mitochondrial markers that differ in their degree of variability. The material and methods used in the article correspond to the tasks set in it. The conclusions obtained correspond to the described results of the genetic analysis. The presentation of the article's material is clear and exhaustive. Despite the generally positive assessment of the manuscript, there are a number of comments and questions about its content.
- The article draws prognostic conclusions, but variable DNA fragments describing only maternal lines are chosen as markers. This approach, despite the unique advantages of mitochondrial markers, as the authors point out, is, in my opinion, reductive in studying the genetic structure of a population. The data obtained in this way does not allow us to fully describe all the processes associated with the genetic differentiation of the studied populations. Therefore, I believe that the authors should discuss this point in the article.
- The importance of the migration activity of the species is repeatedly emphasized in the manuscript, but its features are not briefly described anywhere. Considering that genetic analysis inherently affects only maternal genetic lines, it is not enough to indicate the features of migration activity in individuals of different sexes when discussing the results. Here, it would probably be appropriate to point out the possible assortativity of crossing individuals, which could give advantages in the spread of certain maternal lines in the population.
- The comparative analysis of the variability of Cyt b and CR is somewhat surprising. For example, the phrase "Notably, the D-loop region detected stronger recent gene flow, approximately 60% higher than that observed in Cyt b" is not at all clear. These markers are located in the same ring mtDNA and therefore cannot claim independent variability. In my opinion, we should be talking about a variety of types of combinations of these two markers, and that's all.
- The high haplotypic diversity of the SM population, in my opinion, may be to some extent related to the large sample size. An increase in the sample size of other populations could well lead to an increase in their haplotypic diversity.
- It is not clear why the authors associate the presence of common haplotypes only with migration and do not take into account another process – area fragmentation.
- In Figure 1, it is necessary to identify the studied populations. After studying other maps, it becomes clear, perhaps this is an erroneous opinion, that the populations of SM and JF are closer spatially compared to the remote population of NEM. Given the information obtained in this way, the gene flow described by the authors is not always clear.
- In the Conclusions section, the authors point to the existence of "diverse migration patterns within these populations", but they do not provide information or a description of the models themselves anywhere.
Reviewer 3 Report
Comments and Suggestions for Authors
The authors analyzed the genetic diversity and population genetic structure of tufted deer (Elaphodus cephalophus) in Chongqing, China, using mitochondrial CYTB and D-loop sequences from non-invasively collected 46 samples from three mountains. The study addresses genetic diversity and structure of tufted deer, an under-researched species, filling knowledge gaps for regional conservation. They have also inferred the historical demography of the population. Manuscript fits to the scope of the journal and could be interesting to the readers of Animals.
There are some issues in the manuscript that should be addressed.
• Novelty of this work is moderate as mtDNA-based structure studies have been widely conducted in cervids. Inclusion of nuclear markers (e.g., microsatellites, SNPs) would have strengthened conclusions about gene flow and structure. This limitation should be acknowledged.
• Some important methodological concerns exist about sampling efforts. The sampling strategy is not clear. The manuscript mentions random sampling of fecal samples, but does not clarify. Randomization method in field collection. Clarifying this is critical to avoid pseudoreplication biasing diversity estimates. Further, nothing is mentioned about how resampling of the same individual was avoided (e.g., spatial separation, genotyping individual identification markers).
• Phylogenetic analysis relies on NJ tree. Reliance on NJ trees is suboptimal. Current standards recommend Maximum Likelihood (ML) or Bayesian Inference (BI) approaches for robust haplotype phylogenies. Authors should re-analyze using these methods.
• Figures have too low resolution of haplotype networks and trees limits interpretability. They should be redrawn with higher clarity, consistent color coding, and readable bootstrap supports.
• Table 3 listing individual haplotype compositions is unnecessary given the network figures. Removing such exhaustive tables will streamline the manuscript.
• Terms like “Clod I” appear without definition. Possibly typographical errors needing correction.
• There are several discussion statements in the results. Some results are repeated verbatim in the discussion without deeper biological interpretation. For example, the mechanisms behind the SM population acting as a gene flow hub (ecological corridors, lower human disturbance) should be explored.
• The conclusions recommending corridors for the Jinfo population are appropriate but would benefit from specifying potential routes or ecological feasibility based on landscape analysis.
• The discussion should include how recent expansion signals relate to historical climatic events and what conservation implications does this study report under ongoing habitat fragmentation in Chongqing.
• Several other minor issues exist, especially in the simple summary, there is use of vague terms. Needs rephrasing to avoid general phrases like “animals to move between habitats”; instead, specify “establishing ecological corridors to connect Jinfo and Simian Mountain populations.”
Additional comments are in the annotated PDF. Please revise the manuscript for a better presentation of the study.

Reviewer 4 Report
Comments and Suggestions for Authors
The manuscript, entitled ‘Genetic diversity and population structure of tufted deer (Elahodus cephalophus) in Chongqing, China’, concerns a species of tufted deer living in the mountains of China. Three groups of these animals were studied for the d-loop region and cytochrome B. The genetic diversity and population structure of these animals was then estimated. The work is interesting and, with the exception of two chapters (results and discussion), well written. The analyses were well chosen. Below are my comments.
Abstract
I think the abstract needs to be edited. It should be a summary of the entire manuscript, including a short introduction, research aim, material and methods, results and conclusions.
Introduction
The end of this chapter should be a clearly defined aim of the study. Add a fragment of what exactly was the aim of your analyses.
Materials and methods
L108-L114-I think that placing the sequence in a table would be more readable.
Results:
L139-L141: Delete this sentence.
L161-L167: Increase the font size as in the rest of the text.
L169-L173: I don't understand these sentences. Are you sure you counted these haplotypes correctly?
Figure 2 is a graphical representation of Table 3. I am therefore not sure whether Table 3 is necessary. Consider adding it to the supplementary files.
The results and discussion sections are very similar. They need to be processed. In the results section, present the results. You often use a discussion of why you got the result. Then in the discussion section, you repeat the same facts. Please change that.
Round 2
Reviewer 3 Report
Comments and Suggestions for Authors
Dear authors,
Thank you for revising the manuscript by considering the comments provided in the previous round of review. The manuscript is much improved. There are still some issues in the manuscript which needs careful revision.
- The timescale in the figures a and b of the figure are not appropriate? On what basis you did this calibration for around 20 years? Such scale represents that those mutations which showed the tree topology happened in last 2 decades, which is wrong. Revise it carefully.
- Figure 3 caption, mention specifically what type of network is it, TCS or median joining or else?
- Subheading 3.4 Population Demographic history. No need the word Population.
- There are many typological issues of capitalization of the first letter of the words, especially in the caption.

Reviewer 4 Report
Comments and Suggestions for Authors
Accept in present form.
Author Response
Thank you very much for reviewing our manuscript and for your positive comments. We are honored and thrilled to know that you have recommended it for acceptance. Your affirmation is a great motivation for us to continue our work in this field. Thank you again for your valuable time and support!